# Demonstration Laboratory of Industry 4.0 Retrofitting and Operator 4.0 Solutions: Education towards Industry 5.0

**DOI:** 10.3390/s23010283

**Published:** 2022-12-27

**Authors:** Tamás Ruppert, András Darányi, Tibor Medvegy, Dániel Csereklei, János Abonyi

**Affiliations:** 1ELKH-PE Complex Systems Monitoring Research Group, Department of Process Engineering, University of Pannonia, Egyetem u. 10, POB 158, H-8200 Veszprem, Hungary; 2Sensor Development Research Group, Research Centre for Engineering Sciences, University of Pannonia, Egyetem u. 10, POB 158, H-8200 Veszprem, Hungary

**Keywords:** Operator 4.0, Brownfield Industry 4.0, retrofitting, demonstration laboratory, intelligent space, digital twin

## Abstract

One of the main challenges of Industry 4.0 is how advanced sensors and sensing technologies can be applied through the Internet of Things layers of existing manufacturing. This is the so-called Brownfield Industry 4.0, where the different types and ages of machines and processes need to be digitalized. Smart retrofitting is the umbrella term for solutions to show how we can digitalize manufacturing machines. This problem is critical in the case of solutions to support human workers. The Operator 4.0 concept shows how we can efficiently support workers on the shop floor. The key indicator is the readiness level of a company, and the main bottleneck is the technical knowledge of the employees. This study proposes an education framework and a related Operator 4.0 laboratory that prepares students for the development and application of Industry 5.0 technologies. The concept of intelligent space is proposed as a basis of the educational framework, which can solve the problem of monitoring the stochastic nature of operators in production processes. The components of the intelligent space are detailed through the layers of the IoT in the form of a case study conducted at the laboratory. The applicability of indoor positioning systems is described with the integration of machine-, operator- and environment-based sensor data to obtain real-time information from the shop floor. The digital twin of the laboratory is developed in a discrete event simulator, which integrates the data from the shop floor and can control the production based on the simulation results. The presented framework can be utilized to design education for the generation of Industry 5.0.

## 1. Introduction

The readiness of manufacturing organizations is an essential element of implementing Industry 4.0 solutions. The Industry 4.0 maturity model [1] details the readiness index of organizations [2] and regions [3] to enter the next generation of the industrial revolution. The key element of the Industry 4.0 strategy [4] is step-by-step development. Smart retrofitting is defined as a process of enhancing existing machinery to provide current information about its status [5] as it is the key element of digital transformation, especially in the case of small and medium-sized enterprises (SMEs) [6]. With smart retrofitting, SMEs can transfer the aspects of Industry 4.0 visions to machines or processes effectively with regard to cost and time [7]. Smart-retrofitting-based development is also an enabler for Industry 5.0 solutions [8]. Since SMEs have limited resources, smart retrofitting offers a good solution because it is a fast and cost-effective way of reaching Industry 4.0, thereby improving their productivity and competitiveness [9]. By taking small steps, the continuous development of smart retrofitting fits into the lean philosophy [7]. Many companies struggle with the problem of having legacy production equipment with different types of machines that cannot support Industry 4.0 communication. Given that it is impossible to replace all equipment economically, the digital functionalities of existing manufacturing equipment in systems should be extended and enhanced [10]. The implementation of various applications depends on the sectors and equipment, which makes the process of smart retrofitting difficult. Common points must be identified and general solutions developed [9]. The concept of smart retrofitting goes beyond updating existing machinery. It enables the application of new management and business models [9]. In order to bring about integration, the related engineering knowledge should cover all the different aspects and specialties of different technologies. The engineering knowledge needs of smart retrofitting are also critical bottlenecks, besides the legacy equipment [7]. Furthermore, the maturity model shows that expertise and knowledge are the main reasons for failures.

Recent global issues regarding supply chains [11] and the constant lack of material resources (especially in the electronics industry) are forcing changes to be made in assembly production because the labor costs in developing countries have recently begun to increase. Since de-globalization is expanding [12], companies are considering moving back their manual production from the East (where labor costs are low). The shortage of qualified human workers is a problem of increasingly high priority in the European Union [13]. One of the key elements of Industry 4.0 is production automation and the elimination of low-value-added workplaces. However, we are facing a problem whereby efficient automation is not possible with currently available technologies. Operator 4.0 [14] aims to support human workers with enabling technologies. Currently, since we are at the starting line of Industry 5.0 [15], where humans will be in focus, human operators need to be reinstated in semi-automated manufacturing processes in a highly efficient way. As a result, the enabling technologies for Operator 4.0 solutions [16] need to be applied in a creative, but more generalized way. Wearable sensors (such as smartwatches or virtual reality sets) play a significant role in operators’ monitoring and also in education, as reviewed in [17]. The tools of smart retrofitting are basic resources to monitor and support the operators during production constantly. The level of high-tech applications is unbalanced since larger companies use these technologies more. A small proportion of the firms have developed, as well as standardized innovation and R&D strategies. To strengthen the competitiveness of the entire value chain, all firms, including SMEs along the chain, should be developed, an essential purpose of which is to improve the skills of workers.

The Industry 5.0 strategy of the European Union [18] considers this challenge as it aims to make manufacturing more sustainable, resilient, and human-centered. We need to redesign our academic programs to include these important discussions for these goals. According to the Industry 5.0 goals, engineering education must be transdisciplinary, hands-on, have data fluency, be management-focused, and provide human–machine interaction experiences [19]. The continuously developed new technologies require trans-disciplinary knowledge, as the boundaries between different disciplines are eroding, such as it being difficult to define what mechanical, electrical, or computer engineering is. The new industrial revolutions are necessitating rethinking of what engineering education should be like in the future [19]. “If we do not change the way we teach, 30 years from now, we’re going to be in trouble” [20]. The most critical skill for manufacturing enterprises is change management. They need engineers who can translate the technology into solutions and the solutions into operation in a continuously changing environment [19].

The co-existing Industry 4.0 and 5.0 need to be discussed and clarified [21]. A study already stated that companies did not recognize Industry 5.0 because of the lack of capacity related to Industry 4.0 [22]. The next generation of education needs to address the three main pillars of Industry 5.0: sustainability, human-centered solutions, and the resiliency aspect. The human–machine or human–technology relations will affect the social world [23]. In the meantime, the impacts of the 4th and 5th Industrial Revolution need to be clarified at the universities [24]. The education of sustainability needs to be addressed, and the tasks of the universities are conference and workshop organization paired with companies [25]. Industrial upcycling [26] also needs to be included in this perspective.

A holistic approach should comprehend the entire industrial ecosystem and requires the involvement of managers, policymakers, and universities. Research groups and education play a considerable role in developing a creative environment for innovation. The Triple Helix model [27] describes the needed synergy for developing knowledge infrastructure. According to the model, cooperation among politics (governments, municipalities), business stakeholders, and education (universities, research groups) can be the preconditions for regional industrial development through practical and theoretical knowledge synergy. In addition, this allows local initiatives to be embedded into long-term strategies.

The increasing need for SMEs for external expertise [28] also demonstrates the demand for such educational solutions. The full-scale implementation of Industry 4.0 at the technology level has the most benefits, and these applications should be widely extended. Due to these considerations, the goals of the laboratory are to demonstrate the benefits that a lower-level technological update and digital transformation can offer firms, provide an opportunity to acquire the necessary engineering skills, as well as ensure that the collected best practices and methods can be applied in other environments.

The purpose of the proposed demonstration laboratory is to provide a flexible environment for demonstrating smart retrofitting and Operator 4.0 solutions. According to this aim, the main contributions of the paper are the following:The paper provides an overview of how students should learn advanced IoT sensor technology. According to this, firstly, we provide a review of how the Industry 4.0 and Operator 4.0 retrofitting solutions relate to different aspects of the IoT concept (see Section 2.1);The paper also presents how the IoT concept should be taught and what should be included in the curriculum of the graduate and post-graduate students (see Section 2.2);Subjects and Industry 4.0 instructional materials are defined according to the IoT layers and the ISA-95 standard (see Section 2.3);We propose an education framework that can demonstrate smart retrofitting solutions according to the goals of Industry 5.0;The concept of intelligent space is presented as a key element of monitoring operator activities (see Section 3);We highlight how the education of data science should be fit to the teaching of IoT technology;Based on this concept, a laboratory has been developed that is suitable for simulating industrial case studies and generating data for the validation of the related machine learning algorithms (see Section 4);The details of an education program that can support building the skills needed for building smart retrofitting, IoT sensors, and data science-based Industry 5.0 solutions are presented. We also present the developed digital twin of the laboratory and the strategy for utilizing it to improve the competencies of students (see Section 5);Finally, Section 6 summarizes the main findings and the applicability of the presented framework.

## 2. IoT Layerwise Presentation of Industry 4.0 and Operator 4.0 Retrofitting Solutions and Education Programs

The aim of this section is to provide an overview of how students should learn the advanced IoT-sensor-technology-based development of the Industry 4.0 and Operator 4.0 retrofitting solutions.

### 2.1. IoT Layerwise Presentation of Industry 4.0 and Operator 4.0 Retrofitting Solutions

In this section, we present the Industry 4.0 and Operator 4.0 retrofitting solutions structured according to the five-layer IoT architecture [29], which represents the different aspects of the IoT concept with five layers (see Table 1): *perception* (*L*0), *network* (*L*1), *middleware* (*L*2), *application* (*L*3), and *business* (*L*4).

The *perception* layer (L0) defines the data gathered from the field devices (machines), as well as transforms analog signals into their digital forms, and vice versa. Digitalization starts with instrumentation, namely with the sensor technology and efficient integration of the signals from the sensors. The smart retrofitting is proven to be a generalized technology for the digitalization of the existing manufacturing process. Table 1 shows three primary groups of applications based on the type of information.

The *network* layer L1 includes the communication between devices and data transmission through wired and wireless computer networks. Communication is the bridge between the devices and the IoT architecture. The communication is defined as a direct TCP or UDP/IP stack, as well as a link between a local area network (LAN) and a wide area network (WAN). The IoT system presents messaging protocols for seamless data sharing. Latency is one of the most significant communication issues when multiple devices try to connect, causing the system to delay the procedure. Edge computing is a unique solution enabling systems to process and analyze information close to its source. The Fifth Generation of mobile networks (5G) applies edge computing as a standard [30]. An ad hoc network is mostly used for machine monitoring, where many possibilities to utilize the flexibility of wireless networks are available [31]. Many different technologies are at our disposal to provide the network infrastructure [32] of traceability. The two main groups are radio-based and non-radio-based techniques. The most common technologies are ultra-wideband (UWB) [33], WiFi [34] and Bluetooth Low Energy (BLE) [35]. Information flow in the manufacturing industries is based on the L1 and L2 layers.

The *middleware* layer (L2) covers the information storage and exchange, data management, notifications, and search engines. In the *middleware* layer, data pre-processing is applied to the decision unit. The middleware consists of two main stages. First is the data accumulation, where professionals must prioritize the essential data from the enormous streams. Secondly, the data abstraction collects all the data from IoT and non-IoT systems (enterprise resource planning (ERP)). Data virtualization is used to make data accessible from a single location. Finally, the raw data are managed in multiple forms. According to ISA-95 and the IIoT, process information systems describe the hierarchy. The sensor data and indoor positions are analyzed to prove the information for the application and business layers. Edge computing is applied for machine monitoring [36] because of the better response times that result, as well as the lower bandwidth load. Edge computing draws computation and data storage closer to the machines as a distributed computing system. The position calculation is based on the measurement of anchors using different techniques, e.g., time difference of arrival (TDoA) [37], time of arrival (ToA) [38], angle of arrival (AoA) [39], and received signal strength (RSS) [40].

The *application* layer (L3) focuses on further data analysis to gather business intelligence and user interface design. At the *application layer*, the main KPIs are calculated for the machines based on the sensor data [41] and the changeovers are analyzed [42]. The ML-based techniques are mostly used for fault diagnosis [43] as real-time condition monitoring. Spaghetti diagrams are re-invested based on the IPS data [44] during Lean 4.0 [45,46]. Furthermore, the placed-based KPIs are the basis for the decision systems [47].

The *business* layer (L4) refers to the application of the IoT for business purposes. The preprocessing and analysis of IoT data are valuable only if they apply to business planning and strategies. A discrete event simulation (DES) is proposed to provide decision support for the *business layer* in Section 5. On the business planning and strategy level, real-time machine states are essential for production control [48]. Preventive maintenance is available thanks to real-time condition monitoring. The real-time digital twin [47] is based on traceability information, while process mining algorithms are applied to discover the process flow from the indoor positioning data [49].

Traceability is used in the ISO9000/BS5750 quality procedures, addressing the ability to retrace steps and verify the occurrence of certain events. The purpose of tracing is to monitor the transition between events within a manufacturing system, which is subject to the effects of uncertainty and complexity, so that managers are informed about the current state of the system [50]. Traceability is proven to trace all processes, as well as a record of when, where, and by whom the product was produced. The IPS is proposed to achieve full traceability, described in this section. We propose two different information sources to monitor machines. The stages of machines are about their usage, where the different states are discovered, e.g., working, idle, changeover, quality issues, amongst many others. Machine condition monitoring is a well-known application primarily based on vibration measurements and where unexpected behaviors are recognized [51]. At the L0 level, we propose Brownfield I4.0 solutions to gather the information. The states of the machines are determined to gain a clear picture of usage based on the following:Andon lights, where the color of the analog andon lights (which are not connected to the industrial network) can be identified with an RGB sensor;Accelerometers, where the actual state of the machines can be measured based on their vibrations (idle, changeover, machining);Power consumption measurements, where the usage of the machines can be measured directly.

It is possible to identify abnormalities based on anomalies concerning the energy consumption of the machines. However, the vibration sensors are mainly used in the L0 layer for the purposes of machine condition monitoring. Information is available about the complete production process based on traceability, including transitions between the monitored machines. IPS- [44], RFID- [52], or camera-based monitoring systems [53] are widely used to trace manufacturing processes fully.

### 2.2. IoT Education for Industry 4.0

The concept of Industry 4.0 covers advanced and efficient manufacturing systems based on smart technologies [54]. One of its central pillars is the IoT [55]. The growing role of IoT technology in industry and business is placing demands on educational institutes to develop IoT-relevant skills for students. The consensus is that one of the biggest obstacles to adopting IoT technology is the need for more professionals. Talent demand is expected to outstrip supply several times over in the future, and the number of vacancies in the sector could reach into the millions [56,57]. The IoT-driven microelectronics advancements [58] and, with it, the growth of the needed engineering knowledge [59] can be described by Moore’s law [60], i.e., they double every year. Higher education needs to keep pace with this process, but it faces several challenges.

#### 2.2.1. Challenges in IoT Education

The rapid development of technology brings difficulties to education. IoT engineering students have to prepare for future problems and technologies that do not exist yet [61]. The tools and equipment used during training will likely become obsolete later. The application area is also permanently widened. Instructional materials and labor facilities must be updated regularly, and instructors should be retrained frequently [56]. There is a lack of uniformity in the industry standards and technologies used, which leads to a lack of consensus on the competencies required. Therefore, the educational focus should be on principles and patterns rather than specific technologies [62]. Keeping appropriate equipment also requires money [63]. The IoT concept needs to be viewed from an interdisciplinary perspective, from hardware technology through networks to data processing and business applications. It should integrate expertise from different areas such as sensor technology, cloud computing, and cybersecurity. It requires various engineering and computer science skills, which challenges educational institutes. There needs to be more than an electrical engineering or computer science department to teach the IoT. It is a problem that the IoT is not taught in an integrated way by various departments, such as computer science, electrical and computer engineering, information science, software engineering, information technology security, and information systems [62]. Another argument is that special IoT departments would be needed because departments “borrow” teachers from each other [56]. Other challenges originate from the interdisciplinary nature of the IoT. The different backgrounds of students can lead to difficulties [63,64]. An engineering student has different prior knowledge than an informatics student. An engineering student is less familiar with enterprise information systems architecture, information and communication technologies, or visualization technologies. On the other hand, they have a better understanding of the operation of sensors and the concepts related to production or maintenance. A student coming from a computer science background is expected to understand networks and have programming skills, but he/she does not know the physical aspects of industrial CPS systems. The experience of a collaborative lab has confirmed these concerns for IoT education [65].

#### 2.2.2. Overview of IoT Courses

Without a uniform curriculum, different institutes specialized in different areas teach the IoT with different approaches. Consequently, there is no coherence among different IoT courses [57]. The instruction materials largely depend on the knowledge of teachers, resulting in an abundance of source materials in some specific technology and a scantiness in others [66]. IoT curriculum was described as an “unsystematic patchwork” [56]. The curriculum was classified into full coverage and biased coverage mode. In the case of the former, the subjects are inherited from IT and engineering courses and extended with some IoT-specific subjects. Although this curriculum covers the whole concept, it lacks coherence because the subjects are poorly related. The other mode focuses on some particular aspects of the IoT, which develops a more profound knowledge in those areas, but does not cover the whole IoT concept. Another aspect in which the educational approaches differ is that some courses emphasize the “Internet” side of the IoT, where they concentrate on networks and protocols more, while other courses emphasize the “Things” side of the IoT, in which the hardware elements, such as sensors and controllers, receive most of the attention [67]. There is a third distinction, according to which some courses focus more on technology and cover only a narrow area of possible applications. On the other hand, some courses concentrate more on the application and less on the technology that makes it possible [68]. Poorly designed curricula cause discrepancies between the competencies developed by universities and those expected by industry.

#### 2.2.3. Frameworks for Curriculum Development

Frameworks for curriculum development have been proposed by many. One suggestion is that specific training objectives must be formulated by the expectations of the industry and a convergent curriculum built based on the scientific structure of the IoT concept [56]. The described model structure is made up of four layers, which cover the fundamental aspects of the IoT (*sensing* and collecting data from the environment, data *transmission* through a communication network, information *processing*, *application*) and two supporting technologies, *cybersecurity* and *algorithms*. Another approach is that a curriculum should be built based on the underlying knowledge structure [57]. According to one dimension of the model structure, the three knowledge fields related to the IoT concept, general, professional, and management knowledge, should be covered, and according to the other dimension, the individual subjects are organized around the layers of an IoT architecture model they propose (*perception*, *network*, *supporting*, and *application*). Content for an IoT course starting at Western New England University was designed [67]. The curriculum was organized according to the learning objectives, which indicate the competencies that the students should acquire. These learning objectives are knowledge in areas of fundamental IoT concepts, network protocols, single-board computer usage, programming, and IoT system design and building. It is suggested that the education of necessary skills should be centered on the core technologies of the IoT, embedded computing, sensors, networking, and cloud computing, during curriculum development [68]. In Industry 4.0 engineering education, the curriculum design should not start from specialized knowledge and move towards functionality, but vice versa [59]. The functional domains of Industry 4.0 are defined as designing, modularity, interoperability, virtualization, decentralization, real-time capability, and service orientation. The necessary specialized disciplines from different engineering and scientific fields, such as cloud computing or autonomous robotics, are assigned to these functional domains in an interdisciplinary way. Although the core elements of these curriculum models differ in name and number, there is a substantial overlap in their underlying principles: the competencies to be developed and educational objectives should be organized according to the functionalities required for IoT implementation and application. Based on this thought, a brief summary is presented below, where the necessary technologies and skills are enumerated based on the five-layer IoT architecture [29].

#### 2.2.4. Education Content Summary Based on the Five-Layer IoT Architect

The *perception* layer is realized by sensors and microcomputers. In an educational context, mainly low-cost sensors are used, such as environmental sensors (e.g., temperature, light, motion) and RFID. In addition, biomedical sensors such as heart rate, pulse oximeter, and gas/chemical sensors are sometimes used. Low-cost single-board computers, such as the Arduino and Raspberry Pi, are often the chosen hardware platforms in education projects. Expected competencies based on this layer are understanding sensor operation principles, electrical circuits, signal processing, processing sensor data using foundational programming structures, understanding controllers and operation systems, and the ability to design human–computer interaction.

In the *network* layer, the most frequently used technologies in IoT education are WiFi and Ethernet protocols, which are easy to integrate with the above-mentioned single-board devices. Other less frequently used communication technologies are Bluetooth and Zigbee. Education on this layer should focus on developing an understanding of computer network technologies and design principles and knowledge of network standards and protocols, socket API programming, and cybersecurity.

The curriculum of the *middleware* level uses cloud and web services, such as Amazon Web Services (AWS), Microsoft Azure, IBM Cloud, Google Cloud, and IFTTT. Cloud providers often offer integrated data analytics and visualization services that can be used in IoT curricula (for example, energy monitoring). Outside general cloud service providers, IoT-specific cloud providers are used (e.g., GAIA, AdaFruit.io). The middleware-related learning objectives are designing and implementing the cloud architecture for data management, database management, knowledge of MQTT, configuring cloud-based services for notifications and messages, and designing a dashboard to visualize relevant data.

*Application*-layer-related education frequently uses LED interfaces because of the low cost and easy operation. For visualization and data analytics, web dashboards are commonly used. There are also mobile, LCD, and voice interfaces. The expected skills are knowledge of machine learning (ML) and artificial intelligence (AI) and programming skills (such as Python and JavaScript).

The *business* layer refers to the application of the IoT for business purposes. Several IoT industries have developed, such as smart home and smart logistics. Concerning Industry 4.0, one should mainly consider optimizing production and business processes, digital twins, and collaborative space. The purpose of education at this level is for students to understand the given area.

#### 2.2.5. Application-Oriented Educational Approach in IoT, Industry 4.0, and 5.0 Curriculum

For students to understand the complexity of the IoT applications arising from multidisciplinarity, educational approaches other than the traditional ones are needed [65]. In addition to being multidisciplinary, the IoT is also technology-based, so education aims to help students link theory with application. This can be achieved through different approaches. Industry 4.0 is a top-down approach with a rigid technological structure. On the one hand, this can make continuous improvement, which is in line with the traditional kaizen mentality, difficult. Conversely, some engineers and technicians need help to keep up with digitalization. In order to teach and implement the Industry 5.0 concept, education must pay particular attention to developing engineers’ skills with the right tools to navigate comfortably in the world of digitalization and be able to use it to solve specific problems, adapt to changes, implement lean initiatives, and make continuous improvements [69]. An effective IoT education should seek to ensure that the student learns from content, as well as experience. One way of learning from content is through case studies, as many case studies in the literature can be used as a teaching tool and demonstration for different applications. Traditional lectures also support this kind of learning. The typical form of experiential learning is project- or problem-based learning, where students look for a real-life solution to a practical problem and experience success or failure as a result of their work [62]. The students can apply and develop their skills with a “learning by doing” experience. The project must be comprehensive, covering all diverse aspects (layers) of the IoT, from environment sensing to application, helping students develop their creativity. The project must have a well-defined outcome with a final experience. It is worth doing a project with sub-projects over several semesters, where proper documentation becomes essential. The extent to which students can apply what they have learned is a good indicator of education quality. Collaboration is vital in IoT education because real job scenarios require working on teams with members with different views [65]. In a team of students with different specializations, each student can learn from the others and harness his/her own personal attitudes. This way, students will acquire competencies that will be useful in the work environment, such as communication and leadership. Experiential learning requires platforms and tools so students can learn about IoT technology through hands-on experience. Several IoT courses use IoT laboratories, where students can learn about hardware and software tools. They can test their skills in real application scenarios. The learning factory concept is a physical replica of a factory, where education, experiments, and research can be carried out. Education in this production environment equipped with appropriate tools supports the development of the competencies required by Industry 4.0 [70]. Laboratories are costly to set up, so it makes sense for universities to work with companies to establish them jointly [71]. Today’s technologies enable the establishment of remote and virtual laboratories [72]. IoT tools can be handled in remote labs, and actual processes can be monitored and controlled from a distance through special software. A cost-effective option is to set up virtual laboratories. In virtual laboratories, the experiments and the necessary infrastructure are simulated on computers. Hence, learning is not bound by time and place. Virtual and augmented reality can support this concept. In the future, education will be more personalized and tailored to the individual needs of learners. Professors will increasingly play a mentoring role, and assessments will be based on results [70].

### 2.3. Curriculum for Industry 4.0 and 5.0 Education

In this subsection, we present our proposed curriculum and education supported by the demonstration laboratory. The proposed demonstration laboratory and instructional materials are based on the IoT layers [29] and ISA-95 standard [73]. The IoT-layers-based subjects are shown in Table 2.

The production intelligence subject is based on the *perception* and *network* layers; hence, it familiarizes students with sensors, actuators, and networking technologies. The process information systems subject addresses data preprocessing and integration into the enterprise’s system (*middleware layer*). Data analysis of the *application layer* requires analytical skills, machine learning techniques, and real-time display methods that the data science and machine learning subject prepares students for. The integrated data should be converted into information that can be used for business planning. Digital twins and process simulation introduce the application possibilities of the extracted information, which is supported by the optimization and operations analysis subject. Education must cover the pillars of the Industry 4.0 strategy and the readiness assessment. The agile project and quality management of Industry 4.0 solutions and solutions development for Industry 4.0 subjects ensure the required knowledge. The offered possibilities provided by the laboratory with regard to the aforementioned instructional materials are also summarized. Instrumentation and production traceability are key components. An IPS facilitates full traceability, while a brownfield sensor technology enables machine status and condition monitoring. Different communication channels and an edge computing system are provided. These offer a basis to teach the necessary knowledge and create possibilities to calculate KPIs, as well as conduct production-related analyses and planning.

The training objectives developed are in line with the top-10 most-important skills for the future identified by the World Manufacturing Forum [74]. The instructional material includes the development of the necessary technical knowledge and cybersecurity concerns. In data analysis and machine learning, different algorithms and appropriate tools (Python, KNIME) are taught to develop the right data analysis skills. The teaching method is based on case studies and problems, supporting creative problem solving and the ability to deal with complex and complex tasks. Students work in teams, promoting collaboration and communication skills development. These educational approaches are supported by the necessary infrastructure in the laboratory. The sensor technology in the laboratory that is designed to support the education and demonstration of Brownfield Industry 4.0 will be presented in the following sections.

## 3. Intelligent-Space-Based Education of IoT Tools

In this section, the main elements of the proposed demonstration laboratory and related training elements are described. Figure 1 shows the developed infrastructure with the layers of the IoT. This infrastructure guides the students step-by-step through a general digitalization process and makes a clear scope for it. In the following, we will explain how the intelligent space can support the students and show some example solutions for all IoT layers.

Firstly, the intelligent space concept [75] is proposed with extensions to prove a well-designed framework for the demonstration laboratory. Sensors and cameras can recognize, as well as perceive the processes and actions; moreover, automated responses in real-time can be triggered or information provided by visual and voice signals to the operators. Therefore, a back-and-forth connection can be established between operators and the system, which is the basis for the real-time connection between the shop floor and the digital twin [47]. The IoT and the intelligent space, two similar, but not equivalent concepts [76], deal with distributed devices and represent the distributed networks of devices, respectively. The requirements of the intelligent space are: (1) modularity and adaptability, whereby the system can recognize the changes and, based on this recognition, reorganize itself; (2) that the newly added subsystems can be easily integrated into the system, given that positioning data are the key information required for traceability [77] and digitalization [78]. Therefore, localization is the crucial component of the intelligent space [75]; the IPS locates the position of an item anywhere in a defined space in real-time [44]. The IPS is proposed for the sensor network of the collaborative intelligent space and provides data with regard to the movements of material supply, resources, and the workforce [79].

The IPS is installed to track assets, workers, and logistics vehicles. The structure of the IPS is detailed in the following paragraphs along with the educational possibilities. Industry 4.0 retrofitting sensor solutions are developed to gather the data at the *perception layer*. The IPS uses the UWB technology, while the developed and applied sensors communicate via Bluetooth and WiFi at the *network layer*. An automatic guided vehicle (AGV) developed by us aims to apply a developer tool for many aspects. The extension of the intelligent space considers the *middleware* and *application* layers, which are described in Section 4 and Section 5, respectively.

The IPS is one of the key elements of the proposed intelligent space [16], and it is applied in many cases in manufacturing [44]. The IPS enables full traceability on the shop floor, the information from which can revolutionize the digital lean concept [45]. The IPS is applied in the laboratory, facilitating real-time position tracking of objects, e.g., AGVs, workpieces, and operators. The system structure consists of the following elements: a central unit, anchors, and tags. The laboratory aims to facilitate the application and development of indoor position data processing algorithms, as well as allow the instruction of the signal- and data-processing position-based conditions and state prediction. Application-oriented education is available thanks to the installed IPS with the detailed and user-friendly interface (see Figure 2).

During the digital twin development (see Section 5) of the laboratory, it became necessary to simulate the participants or the events of the production steps separately from the whole process. For this task, an AGV was sought, a tool that already exists in an industrial environment. When simulating these events and participants, complex forms of movement needed to be performed, for which purpose an industrial AGV would be impractical. Therefore, we created a custom-built AGV, which is shown in Figure 3, and we were able to modify its firmware to suit the behavior required. The AGV built for the laboratory was made with an omnidirectional drive equipped with so-called Mechanum wheels to improve its maneuverability and allow it to drive on narrow roads. As a result, our vehicle has become able to simulate the movement of workers, as well as the position and usage of tracked tools during work processes. For this feature, a use case will be shown in Section 5. In addition, given its nature, it was able to perform the typical functions of an industrial AGV.

The vehicle was equipped with several independent positioning options to validate the IPS data. The AGV was able to orient itself in the laboratory using a line-tracking sensor and a boundary track with fixed positions on known coordinates. It could position itself between the fixed points defined by the track using the encoders on its wheels. In addition, an independent laser radar (Lidar) was installed on the AGV. With the help of the data provided by the Lidar, the vehicle could map its surroundings and orient itself regardless of the boundary track using the simultaneous localization and mapping (SLAM) method. The AGV helped us design experiments using the positioning system. Since the exact coordinates of fixed routes and stops are known, it can be used to generate labeled position data independently of the IPS. The labeled data can validate the indoor position data and the developed application-oriented algorithms shown in Section 4.

We developed a flexible and straightforward sensor solution to measure the main statuses of the different types of machines and the working conditions of the operators. The brownfield terminology in Industry 4.0 means that existing manufacturing needs to be digitized. The problem with this procedure is the different types of machines and processes on the same shop floor. For these tasks, we designed a mobile and stationary data acquisition (DAQ) device, as well as the AGV shown above (see Table 3). The wireless mobile DAQ device is based on a low-power microcontroller (see Figure 4a). In addition, such a unit includes a low-power triaxial accelerometer, a microphone, a thermometer, as well as a humidity and dust sensor. We designed an easily reproducible 3D-printed case with a magnetic mounting option for the completed units. Wireless communication takes place via a BLE module. Data streamed by the unit are received by a server machine and saved in a PostgreSQL database for processing. The resulting unit is mobile, inexpensive, and easy to install. It can be mounted on machines or carried by workers or AGVs and can measure the condition of the machines, as well as the working conditions on the shop floor. Due to its low power consumption, it can operate from a battery for days, even using high sampling rates. Combined with the IPS, all of these data can be measured in a position-dependent manner. Therefore, environmental maps of the shop floor can be created.

The other type of data acquisition device can be installed on stationary machines and operated from an AC power source for an indefinite period without the need for maintenance (see Figure 4b). This unit is a magnetically mountable triaxial accelerometer connected to a Raspberry Pi and can also be equipped with a thermometer, as well as an RGB sensor for Andon light state detection. WiFi facilitates wireless communication. The data can be forwarded from the device to the server, from which it can also be uploaded to the PostgreSQL database, or the device can save the measured data directly in the database. This unit is suitable for monitoring the condition of stationary-position machines for a lengthy period of time. It is unnecessary to charge its batteries continuously and is easy to install, and its settings can be changed remotely.

Our goal was to develop brownfield sensor solutions to create a demonstration laboratory where the adoption of Industry 4.0 in a factory could be shown and taught. With the sensors developed by us, we are able to collect labeled data about the operators, machines, and environment (Figure 5). From these data, we can determine the working conditions and comfort level of the operators, as well as gain an insight into the current conditions of the machines. From all of the above, we can determine the KPIs of the production process.

## 4. Education on the Development of Real-Time Process-Monitoring Solutions

Practical training should be based on real-world job scenarios where the students work in small teams to find solutions to real brownfield problems (such as status identification on legacy machines) that also occur in an industrial environment. In this way, in addition to simple technology-related skills, the further competencies expected in a work environment, such as creativity, communication, and problem-solving skills, can be developed in a targeted manner. We defined a production process in the demonstration laboratory, where the work- stations represent the different solutions. These stations or modules help the students understand the possibilities of the proposed smart-retrofitting-based digitalization solutions. The designed sensor-network-based pilot problem creates a platform for the students to develop their solution for real-time process monitoring (even in Python, Node-red, or other platforms). The laboratory is a mechanical engineering laboratory, where many different types of machines are found. The laboratory has milling and drilling machines from the 1960s to the 1980s. The demonstration aims to track a production process through several workstations equipped with sensors. Through this process, students can analyze the sensor data from the actual manufacturing machines- moreover, position data is gathered from the movement of the AGV, workers, and products. Thanks to the different types of machines, the applicability of the proposed sensors is proven.

Figure 6 shows the layout, where the blue rectangles represent the machines. The demonstration tracks the route of the semi-finished product over the workstations (which are also monitored) until it acquires its final form, that is the product. The sensors installed on the machines provide us with data about the generated forces, vibrations, achieved speeds, and elapsed times. The extracted raw data will be processed and, then, used for production optimization (Section 5). The demonstration product (see Figure 7) is produced by milling, drilling, and manual finishing at seven workstations (see also Figure 6) as follows:Material storage;Milling of the sides of the material on the cantilever milling machine;Rough lathing;Smooth lathing;Drilling using a pillar drilling machine;Notch making on the cantilever milling machine;Finalization.

A conventional cantilever milling machine (second workstation) is used, suitable for both horizontal and vertical milling. The vertical option is used to machine the current workpiece, and the vertical milling spindle is stationary. The table has a separate drive motor and can feed the machine in the *x* direction. Two universal lathes are used for turning. Roughing is performed on a ZMM c11m/1000 machine. Smoothing is performed on a TOS Trenčin SM16A-type lathe and drilling using a Hitek FP13/0 column drill.

Figure 8 shows the measured vibration and noise data from the ZMM c11m/1000 machine (third workstation). The sensor is placed directly on the machine. Based on the measured data, five different production phases can be identified, which are the different machining steps. This information will be the basis for future analysis (see Section 5), where the identified times will be used for the process simulation. The worked out example shows the students that smart retrofitting solutions can extract valuable information from even old machines with simple, inexpensive, and easy-to-install tools. They learn to analyze the extracted data in the context of data science, while sensor development is taught in a mechatronic systems course, so students not only learn about the development and installation of sensors, but also directly analyze the outgoing data using data mining methods. Furthermore, the students also learn how to tailor a solution to a specific machine by analyzing the vibration data. Once the necessary information has been extracted from a station, it must be managed in an integrated manner concerning the entire production. The IPS can support the monitoring of the entire production process.

The IPS tracks the production flow. In the laboratory, several tags measure the real-time position of the AGV, workers, and products. The IPS measures the positions of tags with a specific frequency with an accuracy of half a meter. The laboratory is considered to be a non-line-of-sight (NLOS) environment, where covering and interfering effects prevail. As a result, the position data recorded at a specific location are scattered around the location with a certain degree of variance depending on the accuracy of the IPS at that specific location. Therefore, the position data must be processed. Bayesian filters (for example, Kalman filters) [80] can help to filter the position data. Clustering algorithms are suitable techniques for the identification of relevant places or trajectories.

To generate labeled position data, we designed a test using the AGV equipped with a tag. The AGV was programmed to follow a predefined route that contained five stops with known coordinates. These stops simulate workstations, and the AGV spent 10 min at each. The position tag contains an accelerometer that has energy-saving purposes. The sampling frequency increases when the sensor senses motion or vibration, and the sampling rate decreases when the tag is inactive. Given that the tag is usually in an active state at a workstation during the working process, we had to maintain the active sampling frequency at the stops to imitate workstations. This was achieved by a 2–5-degree back-and-forth continuous rotation. Due to the uncertainty of the measurements, the position data from different stops generate distinct distributions.

Figure 9a shows the route and the raw position data of the AGV. These distributions were identified by the Gaussian mixture model (GMM) clustering algorithm, and the created position clusters represent the stops. The model assigns a probability to each data point, allowing anomalies and outliers to be detected. We can filter out data points with lower probabilities than the threshold values. Figure 9b shows the clustering result.

The vibration and noise data are used to obtain a precise picture of the production paired with the indoor position data. The process model is combined with the production parameters to draw up a process simulation for further analysis. In the next section, we introduce an educational tool for process analyses and improvement.

## 5. Teaching the Development of Digital Twins

In the era of engineering education, the curricula must be kept up-to-date. The digital twin concept is a key element of the new industrial revolution. The digital twin can increase motivation for studying when applied correctly since students and teachers can benefit from it [81]. The benefits of using digital twin technology in the *control system design course simulation* are proposed with the detailed models in [81]. Another study proposed a novel robotics teleoperation platform that allows the usage and learning of robotics remotely based on a digital twin with bi-directional data transmission between the physical and digital assets [82]. This section proposes a framework to develop a digital twin through the demonstration laboratory stations and process. Thanks to this simulation model, the students can simulate the process to avoid the “not enough” data problem. Thanks to this, we are preparing our students for real-world scenarios where a huge amount of heterogeneous data must be integrated.

Currently, the production line equipment, workpieces, and machines communicate via the IIoT and form a cyber–physical system (CPS). By cyber–physical production systems (CPPSs) [83], we mean the integration of IT, software technology, as well as mechanical and electronic components, where the basis of communication is the IoT. One of the main features of a CPS is its very high degree of complexity. The design of CPSs is created by networked embedded systems with the help of wireless communication networks. The digital twin is a digital model of a real object that simulates this object in a live setting. This twin is a real-time digital counterpart of an object or a process based on real-time, real-world data measurements [84]. Using the digital twin allows us to analyze the system, as well as identify weaknesses and bottlenecks, the knowledge from which we can optimize and improve our processes. The most significant advantage of using these models is that they provide a near-real-time comprehensive linkage between the physical and digital worlds [85]. One of the main challenges in using digital twins is data collection and creating a thorough database. In order to create the twins, the copy of each process in the digital space must be as accurate as possible, and as much relevant data must be collected as possible. The use of simulations and digital twins is gaining ground, and their requirements are increasing. The use of simulation is primarily recommended where:Analytical methods are not available or are too cumbersome to use;It would be dangerous and/or time-consuming to experiment in a real environment;The system is too complex;Experimenting with the real system would result in a loss of production.

That is why it is essential to support students and professional development with appropriate expertise to map complex systems. However, mapping reality is not easy. Many data are needed to do this as accurately as possible. Proper knowledge of electronics and sensors is essential for data collection and measurements; moreover, the collected data should be processed and stored appropriately. The creation of the 3D environment begins by using design software. Then, the simulation model and the digital twin can be created using the models and the relevant recorded data. This task requires confident proficiency in the use of designer, as well as simulation models. Mapping such complex systems requires different and diverse tools, as well as knowledge, which are summarized in Table 4. Knowledge of database and data engineering is required to compile a database for the data coming from the data collector (Raspberry Pi, Arduino). Several simulation tools are available for process modeling. The Siemens Plant Simulation software was chosen because of its object-oriented feature.

A digital twin of the laboratory is developed to support the development of computational thinking and the systems approach. This system allows students to learn about the world of digital twins and gain a complete overview of what units are needed to build such a complex system. In the created system, they are free to test how changing each parameter would affect the production process, how our decisions are supported by a well-structured simulation, and how each production unit can be controlled with the help of the digital twin. The process-simulation-based digital twin of the Operator 4.0 laboratory is created by the Siemens Plant Simulation software. The Plant Simulation is object-oriented and widely used. Another advantage is that it is free to download and use for students; moreover, several tutorials are available to help them learn how to use it.

The digital twin is based on data collected in the laboratory. Figure 10 shows the simulation model of the demo production in 2D. The model includes the equipment and processes for demo production described in Section 4. IPS data provide an opportunity to determine the position of tools, workpieces, AGVs, and workers in real-time, as well as use them when creating the model and during simulations (layout design, material flow analysis). A custom-developed wireless metering system provides another large group of data collected with relevant data such as vibration data, process times, etc. (see Section 3).

As a first step, we had to reconstruct the environment and machines. To do this, we created a 3D model of the equipment, which was then imported into the plant simulation. The modeled laboratory environment is shown in Figure 11. Using the models, we generated simulation objects, which can be parameterized and operated, thereby creating a functional copy of the machines. They were then placed in the digital space according to the layout of the laboratory. Transportation between workstations was provided by the AGV, which is a line follower, just like in the laboratory (see Section 3). Specific points of the line are assigned to each workstation, which the AGV identifies as stops. It can decide which stop it deems to be its destination based on the given step of the process, that is from which one it can continue, at which one it has to stop, and at which one it must turn around. Working with the operator, the material is transported to the next machine in the process, to a stop, before repeating the process after machining and loading.

The created model can be fully parameterized from an external database. It is possible to read the sensor data collected in the laboratory into the model and change the processing time, downtimes, or changeovers. The material or process flow is also flexible, thanks to the roots between the stations not being predefined. We developed a dynamically changing routing table, which also enables flexible or adaptable changes to the layout. It is possible to change the position coordinates of the machines, so it is even suitable for a layout optimization task. Thanks to the flexibility and the data connection, the data can also be imported in real-time, creating a live connection to actual processes. The model can trigger a process on the production line as a result of an event. For example, it can be calculated that the operator will soon perform the current machining from the data received. The simulation will command the AGV to start at that station and transport the workpiece to the next location. The developed digital twin enables the students and researchers to use it to answer the typical “what if…” questions, e.g., the impact of modifications to process times and failures in productivity. Figure 12 shows the results of one example simulation. It can be noticed that the simulation software is easy to configure to obtain the main KPIs; moreover, thanks to its built-in functions, it is a good tool for further development.

For the development of digital twin technology, the knowledge of professionals should also be broadened. The system created in the laboratory provides an opportunity for students to understand every part of a complex system such as a digital twin. Thanks to their broad vision, professionals can use master optimization techniques and develop systems for optimizations such as digital twins.

## 6. Conclusions

Manufacturing industries, especially SMEs, often face a lack of knowledge and resources required for the development of Industry 4.0 solutions. In such situations, smart retrofitting can offer a fast and cost-effective way to improve productivity and competitiveness. The smart retrofitting of labor-intensive production processes can be effective if it is based on the Operator 4.0 concept.

This work proposed a framework and a demonstration laboratory to show what tools and knowledge are needed for engineers to achieve brownfield digitalization. A production process was specified to demonstrate the developed smart retrofitting solutions through the IoT layers and the necessary skills.

The importance of the education of IoT tools was detailed, and the potential smart retrofitting activities were structured according to the layers. As an example, the details of the developed demonstration laboratory were provided. Furthermore, at the University of Pannonia, a *Data and systems Scientist and Industry 4.0 Solution Development Specialist* postgraduate course has been developed for engineers and project managers from manufacturing industries.

The concept of the intelligent space was presented as an integrating framework of IoT-based developments. The indoor positioning system (IPS) is the essential component of the network layer of the intelligent space. Approaches for smart-retrofitting-focused sensorization were also detailed. Mobile and stationary DAQ units were developed for retrofitting Industry 4.0 sensor solutions. Examples of applications were given that can improve condition monitoring and traceability.

The importance of data analysis was also highlighted. The position data were clustered to improve the accuracy of the IPS and identify the relevant zones. Vibration data were classified to recognize the different working steps of the machines. The details of the data-driven digital twin development were also shown.

Based on the presented concept, we recommend that the smart-retrofitting-based development of Industry 4.0 solutions should strongly focus on:The education of the IoT layers;The proposed concept of intelligent space;The utilization of indoor positioning systems;The goal-oriented sensor development/selection;Data science to demonstrate how it can be used to process sensor data and calculate KPIs;The development of digital twins based on discrete event simulations;The concept of Operator 4.0, which puts the human workers into the focus of the production process and forms the bases of the development of Industry 5.0 solutions.

The presented framework and the provided details of the proposed demonstration laboratory can be utilized to design Industry 4.0 and 5.0 development and education programs, demonstration tools, and smart retrofitting activities. The proposed laboratory and the related course materials cover the main elements of the digital manufacturing concept such as IoT-based sensor networks, the utilization of machine learning algorithms, and digital twins. The whole framework is designed with support of skill development to meet the goals of Industry 5.0 in mind, so that the students can also work on the development of the digital lean concept (or Lean 4.0) and agile manufacturing solutions.

## Figures and Tables

**Figure 1 sensors-23-00283-f001:**
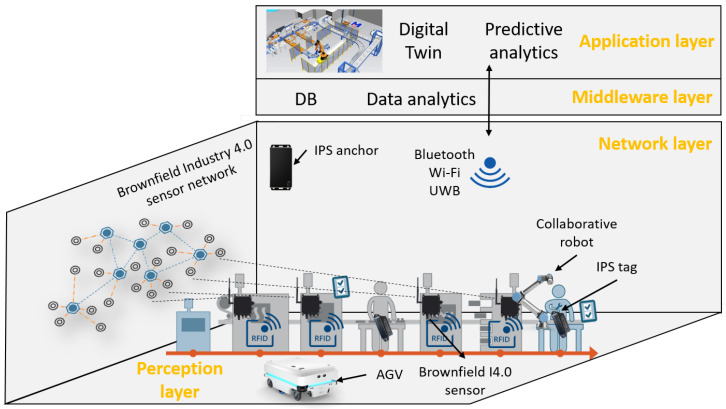
Extended intelligent space to obtain an education framework in a demonstration laboratory.

**Figure 2 sensors-23-00283-f002:**
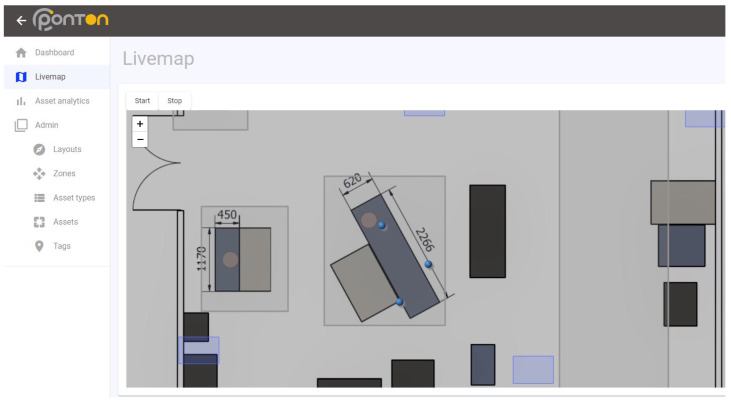
Interface of the IPS with the position of the tags (blue dots) and the layout of the laboratory.

**Figure 3 sensors-23-00283-f003:**
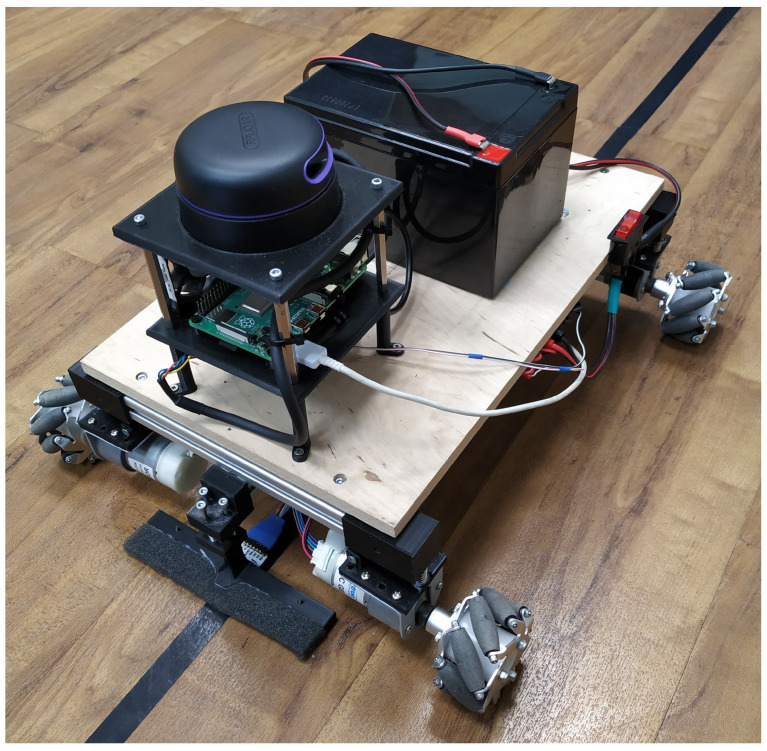
The developed AGV with a Lidar sensor to provide a validation tool for the IPS and environmental mapping.

**Figure 4 sensors-23-00283-f004:**
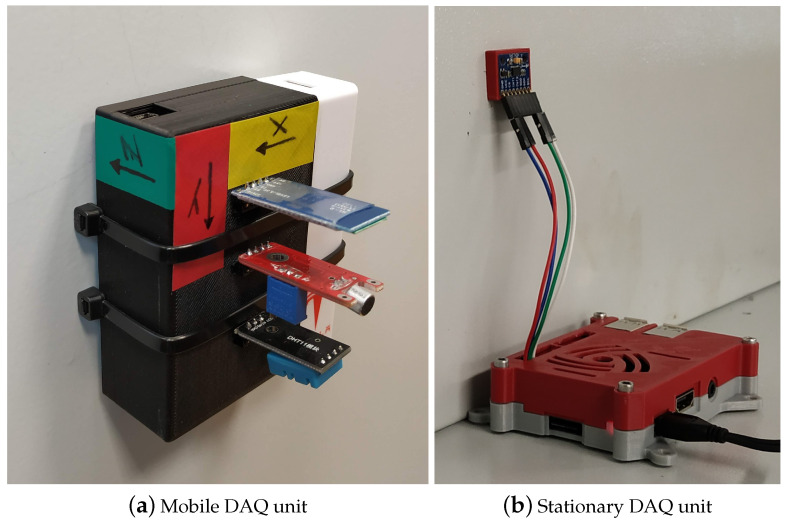
Different types of retrofitting Industry 4.0 sensor solutions.

**Figure 5 sensors-23-00283-f005:**
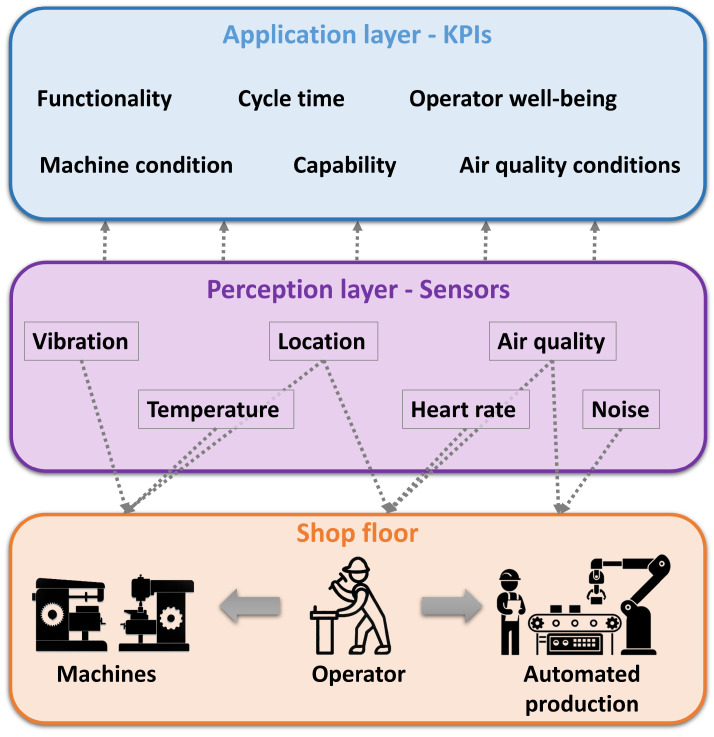
Determining the KPIs of the production process.

**Figure 6 sensors-23-00283-f006:**
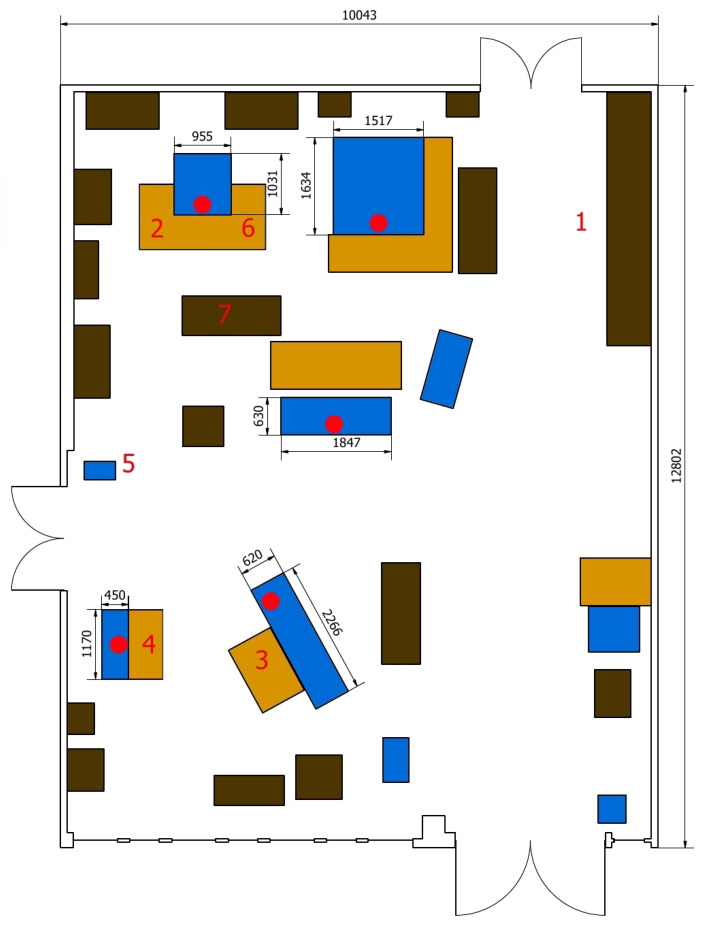
The laboratory layout, where the blue rectangles denote the machines, the red circles stand for the reference points (on the machines) with regard to the IPS, and the numbers represent the production flow.

**Figure 7 sensors-23-00283-f007:**
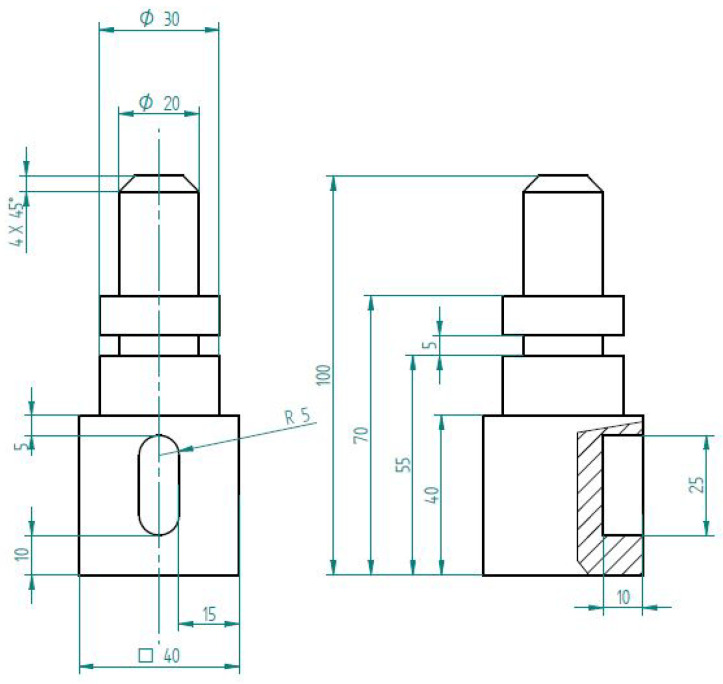
The designed product for demonstration purposes.

**Figure 8 sensors-23-00283-f008:**
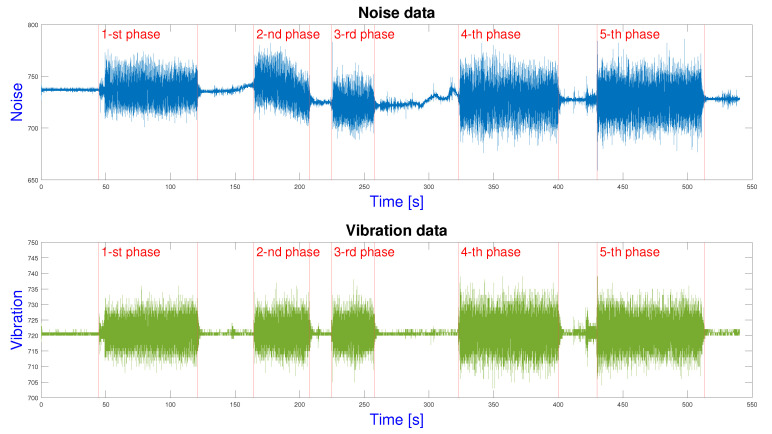
The different phases of machining identified based on the different characteristics of the noise and vibration data from the machine.

**Figure 9 sensors-23-00283-f009:**
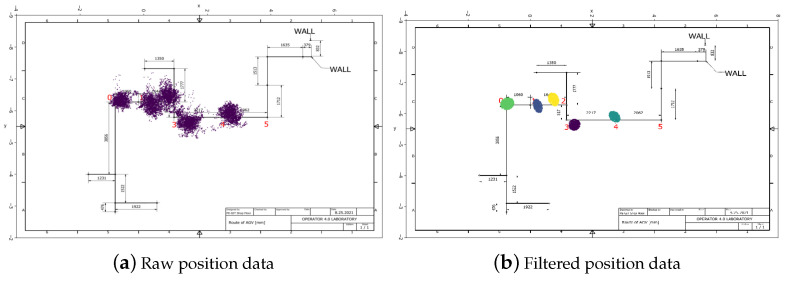
Subfigure (**a**) shows the predefined route and the raw position data. Due to the uncertainty of the measurements, the position data points are scattered and the distributions from two places in close proximity to each other overlap. Subfigure (**b**) shows the result of clustering, as well as of probability filtering, and the well-separated position clusters represent the locations of the workstations.

**Figure 10 sensors-23-00283-f010:**
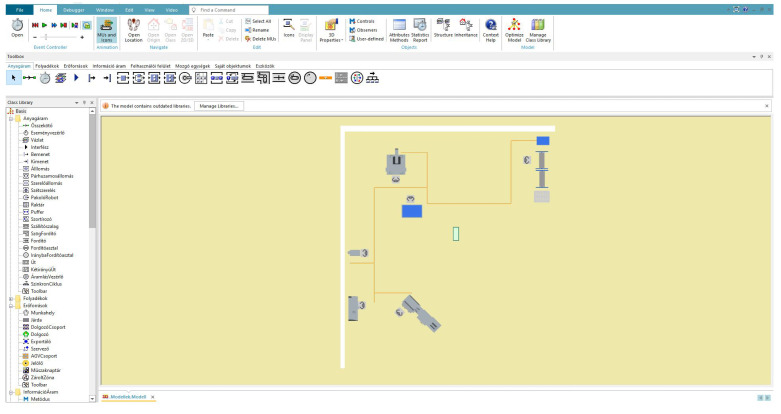
The plant simulation model of the laboratory and the user interface.

**Figure 11 sensors-23-00283-f011:**
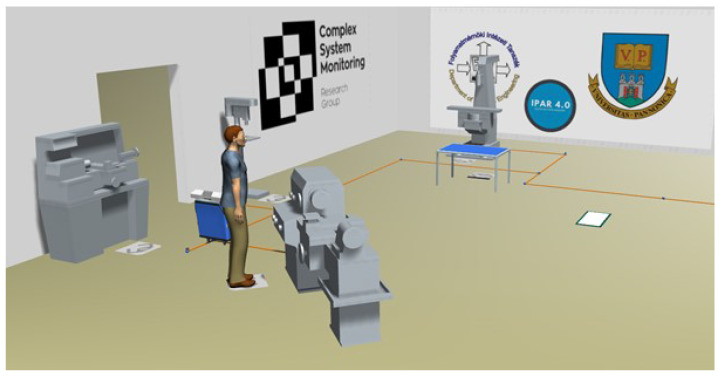
The digital twin of the laboratory.

**Figure 12 sensors-23-00283-f012:**
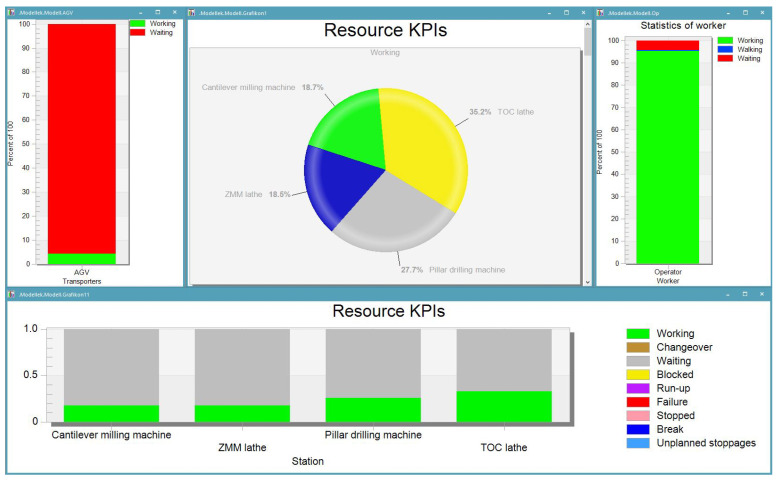
Example for a dashboard of production statistics.

**Table 1 sensors-23-00283-t001:** The proposed applications for Industry 4.0 retrofitting and Operator 4.0 instructional materials based on IoT levels.

IoT ID	Possible Applications		
	Machine States (Usage)	Machine Condition	Traceability
L4	Production control based on real-time machine states	Preventive maintenance	Process mining-based simulation and real-time digital twin
L3	OEE and changeover analyses	Real-time condition monitoring and diagnostics	Spaghetti diagram and location-based KPIs
L2	Edge computing		Position calculation and preprocessing
L1	Ad hoc networks based on Bluetooth, WiFi, or other wireless communication channels		Radio-based: UWB, BLE, ZigBee, WiFi
			Non-radio-based: sound-based, light-based, vision-based
L0	Andon lights, accelerometer sensors, power consumption measurement	Vibration sensors, consumption monitoring	IPS, RFID, camera system

**Table 2 sensors-23-00283-t002:** The proposed subjects for the five levels of the IoT.

IoT Layer IDs	IoT Layers	Layer Description	Proposed Subject
L4	Business layer	Business models for business planning and strategy	Optimization, operations analysis, and artificial intelligence
			Digital twin and process simulation
L3	Application layer	Graphical data representation based on ML and AI algorithms	Data science and machine learning
L2	Middleware layer	Decision unit and data analytics	Process information systems
L1	Network layer	Network technologies and edge computing	Production intelligence
L0	Perception layer	Connect to physical object via sensors and actuators	

**Table 3 sensors-23-00283-t003:** Self-developed sensor solutions.

	Mobile DAQ	Stationary DAQ	AGV
*Condition and state monitoring*	Vibration Temperature Air quality Noise	Vibration Temperature Andon light state	Self-position (track) Self-position (Lidar) Temperature Air quality Noise
*Perception layer sensors*	Accelerometer Thermometer Humidity sensor Dust sensor Microphone	Accelerometer Thermometer RGB sensor	Line-following sensor Motor encoders Lidar Thermometer Humidity sensor Dust sensor Microphone
*Network layer*	BLE	WiFi	BLE or WiFi
*Middleware layer*	Server	Edge computing on a Raspberry Pi or server	Edge computing on a Raspberry Pi

**Table 4 sensors-23-00283-t004:** The required components and tool competencies for process simulation.

Component	Tool	Competence, Knowledge
Process model	Siemens Plant Simulation, Simul8, open-source simulation tools	Modeling skills
Communication protocol	Edge computing, Node-RED	Process informatics
Data-collector, data logger	Raspberry Pi, Arduino	Sensor and electronics knowledge
Database	MySQL, PostgreSQL	Database management skills

## Data Availability

Publicly available datasets were analyzed in this study. This data can be found here: https://www.operator4.com/ (accessed on 2 December 2022).

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
