# Peer review of "Demonstration Laboratory of Industry 4.0 Retrofitting and Operator 4.0 Solutions: Education towards Industry 5.0"

_sensors, 2022, doi:10.3390/s23010283_

Round 1

Reviewer 1 Report

The work presented by the authors is good. The article is well organized and supported by the literature. A real time example is also presented. The article is scientifically sound. It will attract the readers as well as practitioners in the field of Industry 4.0.

Reviewer 2 Report

Dear authors, thank you for your manuscript—here are some comments and recommendations for its improvement by sections.

1. Introduction section.

Please revise the use of the term "digitization", which means bringing analogue information to the digital world. In many cases, I think, you mean the "digitalization" of certain production management operations.

Please provide a formal definition for "retrofitting", or to be more correct in the use of the term in the Industry 4.0 context, "smart retrofitting". Such a definition is missing.

The context described in your introduction is a little bit confusing, mixing the industrial environment with an educational setting. Please rewrite to clarify your context.

Also confusing when reading the introduction section of your paper is its focus on "training students or developing smart retrofitting solutions" or actually "training students in the development of smart retrofitting solutions". It is not clear what is the objective of your paper.

The introduction section seems to mix the paper's motivation with the research setting. Perhaps describing the "intelligent space" and its technologies in a separate section.

2. IoT Education for Industry 4.0 section.

Good explanation of the challenges and opportunities of IoT Education.

The next sections of your paper could be better organized using the three layers of IoT Education: hardware, middleware, and presentation. You jump directly to discuss the presentation layer, but what about the hardware and middleware layers? Perhaps one or two paragraphs about it for each missing layer. This would be appreciated by the reader.

3. IoT layer-wise presentation of Industry 4.0 and Operator 4.0 retrofitting solutions section

This section makes an excellent description of the technological setting for the development of IoT solutions but I'm missing the educational discussion.

4. Intelligent space-based presentation of IoT tools section.

Should the "intelligent space" be described first and then the approach for IoT solutions development discussed in Section 3? The paper's organization makes it a little bit hard to read and follow.

Similar feedback than in Section 3, missing the educational discussion.

5. Education "on" the development of real-time process monitoring solutions section.

Same problem as in Sections 3 and 4, where is the educational discussion? How are you teaching these IoT technologies (e.g. project-based education)? What digital skills students are developing? How they are learning to digitalize a process or smart retrofit a machine tool which is different from just installing an IoT device?

6. Teaching the development of digital twins section.

Same feedback as in Sections 3, 4, and 5. You are missing completely the educational discussion and just focusing on describing the technicalities of the IoT technologies available in your lab. Please provide such missing discussions.

7. Conclusion.

After reading the whole paper, it feels more like a work describing the successful development of smart retrofitting IoT solutions for a brownfield laboratory at a university than a paper with a good balance of discussion between the importance of hands-on IoT Education with a focus on digital skills development for the successful deployment of IoT solutions in brownfield environments based on smart retrofitting strategies. 

The paper needs to clarify its objectives and storyline.

Reviewer 3 Report

Need

corrections to minor methodological errors, Need to discuss contributions more precisely

Reviewer 4 Report

1. The authors must show the contribution and scientific novelty in accordance with the development of an assessment methodology to measure the Industry 4.0 potential and capabilities of firms engaged in manufacturing.

2. The investigators must correlate the Industry 4.0 and 5.0 with the digital transformations in a detailed enumerated manner with proper illustrations.

3. Justify the same that how the authors can approach to control the production parameters in the assembly lines using the proposed methodology which they have mentioned in their study. Take this comment the foremost one.

4. The authors must exhibit to illustrate the hybrid model, using lean or agile manufacturing with the Digital manufacturing concept to meet the goals of Industry 5.0.

5. Which demonstration platform has been employed by the authors of the current work for teaching Industry 5.0? The authors must correlate their results with the previous literary findings to show the novelty.

6. The gap is not clear after contemplating the article in a detailed manner. The authors must show the gap with the research questions and objectives accordingly.

7. The authors must enumerate the provocations of Industry 5.0 for different manufacturing enterprises.

8. The quality of the figures are very awful as the figures are not matching with the standards of the Journal.

9. The authors confirm that whether these figures are original or not, if reused/reproduced from the existing studies, then, show the copyright rights and permissions form.

10. The use of English language is very poor as there are myriads of syntax errors including the English grammatical and typography errors.

11. The main concern with this article is the nature and scope of the work is not matching with the standards of the Journal, even though, there is no correlation with the scope of the Journal. How the Editors have accepted this article for peer-review process is utterly a massive question?

DECISION: REJECTION

Round 2

Reviewer 2 Report

Thank you very much for addressing all comments and recommendations provided. Your paper is ready for publication.

Reviewer 3 Report

Accept with minor english language check